# Comparison of the Characteristics and Outcomes of COVID-19 Patients Treated by a Hospital-at-Home Service in Japan during the Alpha and Delta Waves

**DOI:** 10.3390/jcm11113185

**Published:** 2022-06-02

**Authors:** Ryota Inokuchi, Xueying Jin, Masao Iwagami, Yu Sun, Ayaka Sakamoto, Masatoshi Ishikawa, Nanako Tamiya

**Affiliations:** 1Department of Health Services Research, Faculty of Medicine, University of Tsukuba, 1-1-1 Tenno-dai, Tsukuba 305-8577, Ibaraki, Japan; kinnsetsuei@md.tsukuba.ac.jp (X.J.); iwagami-tky@umin.ac.jp (M.I.); masa.ishikawa1221@gmail.com (M.I.); ntamiya@md.tsukuba.ac.jp (N.T.); 2Health Services Research and Development Center, University of Tsukuba, 1-1-1 Tenno-dai, Tsukuba 305-8577, Ibaraki, Japan; 3Graduate School of Comprehensive Human Sciences, University of Tsukuba, 1-1-1 Tenno-dai, Tsukuba 305-8577, Ibaraki, Japan; happy-go-lucky65@hotmail.co.jp (Y.S.); ayaka.furukawa@gmail.com (A.S.)

**Keywords:** out-of-hours services, emergency department, public health center, after-hours house-call medical service

## Abstract

Coronavirus infections occurred in repeated waves caused by different variants of severe acute respiratory syndrome coronavirus 2 (SARS-CoV-2), with the number of patients increasing during each wave. A private after-hours house-call (AHHC) service provides hospital-at-home (HaH) services to patients in Japan requiring oxygen when hospital beds are in short supply. This retrospective study aimed to compare the characteristics of COVID-19 patients treated by the AHHC service during the COVID-19 waves caused by the Alpha (March–June 2021) and Delta (July–December 2021) SARS-CoV-2 variants. All patients with COVID-19 treated by the AHHC service from March to December 2021 while awaiting hospitalization were included. The data were collected from medical records and follow-up telephone interviews. The AHHC service treated 55 and 273 COVID-19 patients during the Alpha and Delta waves, respectively. The patients treated during the Delta wave were significantly younger than those treated during the Alpha wave (median: 63 years and 47 years, respectively; *p* < 0.001). Disease severity did not differ significantly between the two waves, but the crude case-fatality rate was significantly higher during the Alpha wave (10/55, 18.2%) than during the Delta wave (4/273, 1.4%; *p* < 0.001). The patient characteristics and outcomes differed between the Alpha and Delta waves.

## 1. Introduction

Since the coronavirus disease (COVID-19) pandemic began in December 2019 [1], most countries have experienced repeated waves caused by different strains of severe acute respiratory syndrome coronavirus 2 (SARS-CoV-2), with the number of patients increasing during each wave [2,3]. This led to hospital bed shortages, and many patients could not be admitted to hospitals for oxygen therapy or to receive immediate inpatient care [4].

Pre-COVID-19 studies have suggested that the hospital-at-home (HaH) service, a healthcare modality that administers specialized medical home care to patients for illnesses that normally require hospitalization, can help to reduce the burden of emergency and intensive care departments [5,6]. HaH could be an effective option for COVID-19 patient management, and could reduce the risk of cross-infection in hospitals and help prevent the collapse of the healthcare system [7,8].

In Japan, the number of hospitalized COVID-19 patients rapidly increased in metropolitan areas between March and June 2021 during the Alpha wave. A private after-hours house-call (AHHC) service functioned as an HaH service and provided oxygen treatment for COVID-19 patients in their homes or assisted-living facilities while they were awaiting hospital admission [9]. This service also provided oxygen treatment for COVID-19 patients awaiting hospital admission between July and December 2021 during the wave caused by the Delta variant, which was characterized by increased transmissibility, increased risk of hospitalization, and increased virulence in unvaccinated and partially vaccinated people [10,11]. The case-fatality rate of COVID-19 patients in Japan was 1.9% and 0.4% during the Alpha and Delta waves, respectively [12].

During the Alpha wave, the case-fatality rate of patients treated by the AHHC service was slightly higher than that of hospitalized patients [9]. The subsequent Delta wave led to increased utilization of the AHHC service due to more patients needing oxygen therapy while awaiting hospitalization.

This study aims to compare the characteristics of COVID-19 patients treated by the AHHC service during the Alpha and Delta waves.

## 2. Materials and Methods

### 2.1. Study Design, Patients, and Ethical Considerations

This retrospective, cohort study included all COVID-19 patients treated by the AHHC service between March and June 2021 (Alpha wave) and July and December 2021 (Delta wave). The study was approved by the University of Tsukuba Research Ethics Committee (approval number: 1527). Patient consent was waived owing to the retrospective nature of the study.

### 2.2. Definition of the Severity of COVID-19

According to the Japanese guidelines developed by the Ministry of Health, Labour and Welfare and the National Institute of Infectious Diseases [13], COVID-19 severity is based on respiratory symptoms and classified into four groups: (1) mild, SpO_2_ ≥ 96%, no respiratory symptoms or shortness of breath, coughing alone; (2) moderate I, 93% < SpO_2_ < 96%, shortness of breath or pneumonia findings; (3) moderate II, SpO_2_ ≤ 93%; and 4) severe, admission to an intensive care unit or requirement for mechanical ventilation. Hospital admission was prioritized based on this severity index. This severity grading is similar to that of the US Centers for Disease Control and Prevention COVID-19 treatment guidelines, which classify severity as mild, moderate, severe, or critical [14].

### 2.3. After-Hours House-Call Service

Several developed countries offer AHHC services that deploy doctors to patients’ residences [15,16]. In Japan, a large private AHHC service operates 7 days a week outside regular hospital hours [17,18].

Since the start of the Alpha wave, some regions in Tokyo and Osaka (the first- and second-largest cities in Japan, respectively) requested that AHHC services provided at-home/in-facility treatment to patients with moderate I and moderate II severity who could not be admitted to a hospital; this is the HaH service. After the Delta wave, Saitama and Chiba requested the same service.

After consultation with a public health center, patients with moderate I illness received oxygen and three telephone consultations per day from AHHC; those with moderate II severity received oxygen, dexamethasone, and eight telephone consultations per day. During the Alpha and Delta waves, the Ministry of Health, Labour, and Welfare (MHLW) had not approved antiviral drugs (e.g., favipiravir or remdesivir) for out-of-hospital settings.

### 2.4. Data Sources

The study used anonymized data from the medical records of all COVID-19 patients who used the AHHC service and records of follow-up telephone surveys conducted 30 days after the consultation.

Data on the following patient variables were collected from medical records during the first AHHC service consultation: age, sex, body mass index, smoking history, alcohol consumption, comorbidities (hypertension, hyperlipidemia, diabetes mellitus, cardiac disease, cerebral infarction, liver disease, chronic lung disease, cancer, chronic kidney disease, dementia), time from symptom onset to consultation at the public health center, the severity of COVID-19, vital signs, radiographic findings, and symptoms (shortness of breath or difficulty breathing, fever or chills, fatigue or tiredness, cough, diarrhea, headache, muscle pain or body aches, vomiting, sore throat, change in or loss of taste or smell, nasal congestion or rhinitis, or no symptoms). Data on subsequent hospital admission and mortality were collected during follow-up telephone surveys.

To compare the number of patients treated by the AHHC service and the nationwide number of COVID-19 patients, we used national surveillance data obtained from the MHLW [19].

### 2.5. Statistical Analysis

The baseline characteristics were reported as median and interquartile range (IQR) for continuous variables and as proportions for categorical variables. The characteristics and outcomes of patients who used the service during the Alpha (March–June 2021) and Delta waves (July–December 2021) were compared. The statistical significance of differences between groups was analyzed using the Mann–Whitney U test for continuous variables and the chi-square test or Fisher’s exact test for categorical variables. Analyses were performed using JMP 16.1 statistical software (SAS Institute Inc., Cary, NC, USA). *p* values < 0.05 were considered statistically significant.

## 3. Results

### 3.1. Patient Characteristics

This study included 55 and 273 COVID-19 patients who used the AHHC service during the Alpha and Delta waves, respectively (Table 1). The median age of patients was lower during the Delta wave (47 years, IQR: 38–56.8 years) than during the Alpha wave (63 years, IQR: 49–81 years). In both waves, there were more male than female patients. The prevalence of a smoking history and alcohol consumption was significantly higher during the Delta wave than during the Alpha wave. There was no difference in COVID-19 severity between the two waves (moderate II, 72.7% and 75.1% during the Alpha and Delta waves, respectively; *p* = 0.71); however, the prevalence of comorbidities was higher during the Alpha wave than during the Delta wave, and the crude case-fatality rate was significantly higher during the Alpha wave (18.2%) than during the Delta wave (1.5%, *p* < 0.001). The median interval from symptom onset to consultation was longer during the Delta wave (7 days, IQR: 5–9 days) than during the Alpha wave (6 days, IQR: 2.5–9 days). The proportion of patients who were subsequently admitted to hospital, 65.5% (36/55) and 70.3% (192/273), and the interval from consultation to hospital admission, were similar during the two waves. The other patients recovered with HaH care alone. No patients died during HaH care.

According to the national statistics, the number (Figure 1) and the crude case-fatality rate of COVID-19 patients were 6510/350,398 (1.9%) and 3973/943,478 (0.4%) during the Alpha and Delta waves, respectively.

### 3.2. Vital Signs, Symptoms, and Test Results at the Time of the Consultation

The median patient body temperature and heart rate were higher during the Delta wave than during the Alpha wave, but there were no other significant differences in the vital signs between the patients treated by the AHHC service during the two waves (Table 2).

The major symptoms were shortness of breath or difficulty breathing, fever or chills, and cough during both periods. The proportion of patients with fever or chills, fatigue, diarrhea, muscle pain or body aches, and sore throat was lower during the Delta wave than during the Alpha wave. All patients had symptoms, and portable chest radiography and electrocardiography did not detect any abnormal findings during either period.

## 4. Discussion

### 4.1. Summary of Findings

This study revealed some striking differences in the characteristics and outcomes of COVID-19 patients treated by the AHHC service during the Alpha and Delta waves of the COVID-19 pandemic. A total of 55 and 273 patients used the AHHC service during the Alpha and Delta waves, respectively. Although disease severity was similar between the two waves, the crude case-fatality rate among the patients was significantly lower during the Delta wave than during the Alpha wave. The increased number of patients treated by the AHHC service during the Delta wave was reflective of an increased number of COVID-19 cases nationwide. Among the patients treated by the AHHC service, the prevalence of fever or chills, fatigue, diarrhea, muscle pain or body aches, and sore throat, was lower during the Delta wave than during the Alpha wave. To our knowledge, there are no other reports comparing the characteristics of COVID-19 patients treated by HaH services during different waves of the COVID-19 pandemic.

### 4.2. Case-Fatality Rate

We attribute the young age of patients and low case-fatality rate during the Delta wave to the prioritization of older adults and patients with comorbidities to receive COVID-19 vaccination, which started in mid-February 2021 (just before the start of the Alpha wave) [20,21]. In addition, the high case-fatality rate of patients during the Alpha wave may be because the service catered to old patients during this period. Previous studies have reported the establishment of HaHs during the COVID-19 pandemic [8,22,23,24]; one HaH service treated moderate COVID-19 patients [8,24], whereas others treated old and severe patients [23]. The young age and low case-fatality rate among patients treated by the AHHC service during the Delta wave was reflective of the national trend [25] and is thought to be primarily attributable to the success of vaccination, rather than an improvement in the AHHC service.

In this study, no deaths occurred at home in patients treated by the AHHC service during the study periods. Deaths at home resulting from delayed intervention or delayed discovery of abnormality was a social issue, especially in the elderly during the Alpha wave and the middle-aged during the Delta wave in Japan [26]. This was reportedly due to public health centers being too busy to observe patients at home during the waves [27]. Therefore, HaH care may provide a safe and effective alternative for patients with COVID-19.

### 4.3. Comparison of the Characteristics of the Patients with COVID-19 in Nationwide and the AHHC Service Use

In this study, the high number of patients who used the AHHC as an HaH service during the Delta wave and their young age reflects changes in the nationwide COVID-19 epidemic situation. Therefore, the role of HaH is likely to continue to change in the future as new SARS-CoV-2 variants continue to emerge and as vaccination coverage improves, changing the characteristics of the nationwide epidemic. If a new variant tends to infect older adults or leads to a high case-fatality rate, increasing the number of hospital staff and hospital beds should be prioritized; however, if a new variant is associated with a low case-fatality rate and the number of patients is high, HaH services should be expanded and a system that can provide oxygen treatment at home should be established.

Establishing a coordinated referral system with hospitals is important in anticipation of future surges in the number of COVID-19 cases, and the criteria for treatment by HaH services should adapt to the number of available hospital staff and beds, in order to provide the best available medical care for COVID-19 patients and make optimal use of limited medical resources.

### 4.4. Symptoms at Consultation

This study shows that the proportion of patients with fever or chills, fatigue, diarrhea, muscle pain or body aches, and sore throat was lower during the Delta wave than that during the Alpha wave. Since the beginning of the COVID-19 pandemic, SARS-CoV-2 has mutated, and different variants with varying symptoms, disease manifestations, and disease severity, have emerged. Fever, cough, and loss of sense of smell or taste are the most common symptoms reported among individuals infected with the Alpha variant [28], whereas fever, rhinitis, headache, and sore throat are the most common symptoms reported from the Delta variant [29]. Rhinitis, headache, fatigue, sneezing, and sore throat are the most common symptoms reported in individuals with the Omicron variant [30]. Recent studies have shown that COVID-19 symptoms also vary according to the background (age, sex, comorbidities, and genetic background) of the infected individual [31]. Thus, detailed reporting of symptoms on a country-by-country basis is important for physicians treating patients in out-of-hospital settings.

### 4.5. Limitations

This study has some limitations. First, it focused on a single AHHC service with a limited number of patients, so there is potential for selection bias. However, this AHHC service is the largest HaH service in Japan. Second, the study was conducted over a seven-month period and compared two consecutive waves of the COVID-19 epidemic in Japan. SARS-CoV-2 continues to mutate, and the results would probably have been different if cases from other waves were included. Third, it is not possible to determine the extent to which the differences in patient characteristics and outcomes between waves were attributable to differences in the SARS-CoV-2 variant, increasing vaccination coverage in the population, and improvements in AHHC service delivery. Fourth, since we could not obtain nationwide data on the number of COVID-19 deaths and hospitalized patients stratified by age during the Delta wave, we were unable to compare the case-fatality rate between hospitalized patients and those who were treated by the AHHC service. Fifth, to compare the case-fatality rate of COVID-19 patients treated by the AHHC service between the Alpha and Delta waves, we could not adjust the confounder, because the number of deaths was low. Sixth, we did not retrieve information on patients’ vaccination status; thus, we could not include this as a variable in the study. Finally, there is a possibility of strains overlapping during the end of the Alpha and beginning of the Delta study periods.

## 5. Conclusions

The characteristics of patients treated by the AHHC service changed between the Alpha and Delta waves. The role of HaH can change substantially during a relatively short period; thus, to cooperate with the government, public health center, and HaH, the criteria for referring COVID-19 patients to HaH services should be updated based on patient characteristics.

## Figures and Tables

**Figure 1 jcm-11-03185-f001:**
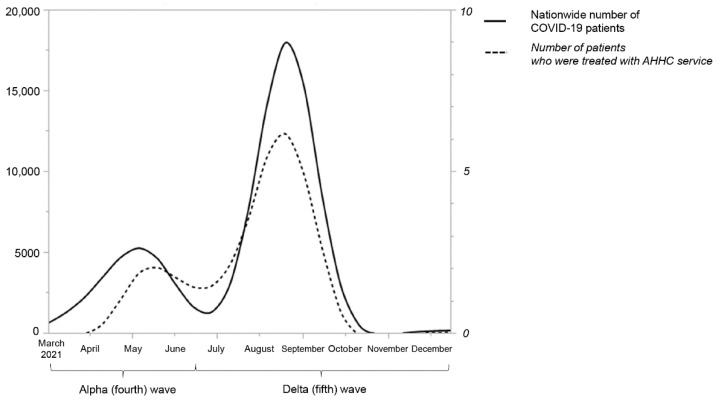
The number of COVID-19 patients treated by the after-hours home-call service and the nationwide number of COVID-19 patients. AHHC—after-hours house-call.

**Table 1 jcm-11-03185-t001:** Characteristics of the COVID-19 patients who received the after-hours home-call service while awaiting hospitalization, according to the wave of the pandemic.

	**Alpha Wave * (*n* = 55)**	**Delta Wave ^†^ (*n* = 273)**	** *p* **
Age, years, median (IQR)	63 (49–81)	47 (38–56.8)	<0.001
Age group, years, *n* (%)			<0.001
0–17	0	2 (0.7)	
18–64	28 (50.9)	235 (86.4)	
65–74	6 (10.9)	16 (5.9)	
≥75	21 (38.2)	19 (7.0)	
Sex, *n* (%)			0.31
Male	35 (63.6)	193 (70.7)	
Female	28 (36.4)	80 (29.3)	
Body mass index, kg/m^2^, median (IQR)	24.3 (22.2–27.7)	24.2 (22.4–27.4)	0.93
Smoking status, *n* (%)			<0.001
Current smoker	2 (3.6)	63 (23.1)	
Ex-smoker	11 (20.0)	7 (2.6)	
Never smoked	32 (58.2)	113 (41.4)	
Unknown	10 (18.2)	90 (33.0)	
Alcohol consumption, *n* (%)			<0.001
Daily	9 (16.4)	64 (23.4)	
Occasional	8 (14.6)	51 (18.7)	
None	30 (54.6)	62 (22.7)	
Unknown	8 (14.6)	96 (35.2)	
Comorbidities, *n* (%)			
Hypertension	25 (45.5)	31 (11.4)	<0.001
Hyperlipidemia	2 (3.6)	12 (4.4)	0.80
Diabetes mellitus	8 (14.6)	18 (6.6)	0.06
Cardiac disease	10 (18.2)	10 (3.7)	<0.001
Cerebral infarction	6 (10.9)	4 (1.5)	0.002
Liver disease	1 (1.8)	3 (1.1)	0.52
Chronic lung disease	1 (1.8)	0	0.17
Cancer	6 (10.9)	8 (2.9)	0.02
Chronic kidney disease	2 (3.6)	2 (0.7)	0.13
Dementia	4 (7.3)	0	<0.001
Days from symptom onset to the consultation, median (IQR)	6 (2.5–9)	7 (5–9)	0.02
Hospital admission, *n* (%)	36 (65.5)	192 (70.3)	0.48
Days from consultation to hospital admission, median (IQR)	2.5 (1.3–6.5)	3 (2–5)	0.77
COVID-19 severity			0.71
Moderate I	15 (27.3)	68 (24.9)	
Moderate II	40 (72.7)	205 (75.1)	
Death, *n* (%)	10 (18.2)	4 (1.5)	<0.001
Death by age group, years, *n* (%)			
0–17	0	0	
18–64	4 (7.3)	4 (1.5)	
65–74	1 (1.8)	0	
≥75	5 (9.1)	0	

* The Alpha wave occurred between March 2022 and June 2022. ^†^ The Delta wave lasted from July 2021 until December 2021. IQR—interquartile range; COVID-19—coronavirus disease 2019.

**Table 2 jcm-11-03185-t002:** Patient vital signs and symptoms at the time of the consultation, according to the wave of the pandemic.

	**Alpha Wave * (*n* = 55)**	**Delta Wave ^†^ (*n* = 273)**	** *p* **
Vital signs			
Temperature (°C), median (IQR)	37.2 (36.6–37.9)	37.9 (37.3–38.5)	<0.001
Heart rate, median (IQR)	87 (80–92)	90 (84–102)	0.001
Mean arterial pressure, median (IQR)	95.3 (81.7–103.2)	89.8 (81.8–96.7)	0.13
SpO_2_ (%) under room air, median (IQR)	92 (90–94)	92 (90.5–93)	0.99
Nasal cannula oxygen support (L/min)	1.5 (0.5–3)	2 (1.5–3)	0.01
SpO_2_ (%) 1 h after oxygenation, median (IQR)	95 (94–97)	96 (95–97)	0.06
Japan Coma Scale, *n* (%)			0.07
Clear	48 (87.3)	257 (94.1)	
Delirium	7 (12.7)	15 (5.5)	
Symptoms, *n* (%)			
Shortness of breath or difficulty breathing	39 (70.9)	207 (75.8)	0.45
Fever or feeling feverish/having chills	43 (78.2)	171 (62.6)	0.02
Fatigue (tiredness)	39 (70.9)	77 (28.2)	<0.001
Cough	34 (61.8)	190 (69.6)	0.26
Diarrhea	15 (27.3)	28 (10.3)	0.002
Muscle pain or body aches	12 (21.8)	22 (8.1)	0.005
Headache	11 (20.0)	36 (13.2)	0.21
Vomiting	11 (20.0)	39 (14.3)	0.30
Sore throat	10 (18.2)	19 (7.0)	0.01
Change in or loss of taste or smell	8 (14.6)	18 (6.6)	0.06
Runny or stuffy nose	2 (3.6)	13 (4.8)	>0.99
Portable chest X-ray finding, *n* (%)			
No abnormality	17 (100) ^‡^	38 (100) ^‡^	
Electrocardiogram, *n* (%)			
No abnormality	17 (100) ^‡^	38 (100) ^‡^	

* The Alpha wave occurred between March 2021 and June 2021. ^†^ The Delta wave lasted from July 2021 until December 2021. ^‡^ The percentages were based on the number of participants who underwent portable radiography and electrocardiography. IQR—interquartile range.

## Data Availability

The data presented in this study are available on request from the corresponding author.

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
