# Peer review of "Comparison of the Characteristics and Outcomes of COVID-19 Patients Treated by a Hospital-at-Home Service in Japan during the Alpha and Delta Waves"

_jcm, 2022, doi:10.3390/jcm11113185_

Round 1
Reviewer 1 Report
Dear authors. I have enjoyed reading your manuscript. I consider a very interesting and relevant work. I send some comments and improvements.
TITLE
Adequate, reports the correctness of the study.
ABSTRACT
The first sentence is previewing the results of the study. It is suggested to replace with a sentence that gives context to the study.
INTRODUCTION
In the first paragraph, a context of the variants and the historical succession of the waves in the covid should be given. I consider this reference very relevant (not mine): Pecho-Silva S, Barboza JJ, Navarro-Solsol AC, Rodriguez-Morales AJ, Bonilla-Aldana K, Panduro-Correa V. SARS-CoV-2 Mutations and Variants: what do we know so far? Microbes Infect Chemother. 2021; 1: e1256.
METHODS
Mention the dates of data collection.
Inclusion and exclusion criteria.
Were normality tests performed?
Sample size or statistical power.
RESULTS
Very well presented.
DISCUSSION
The possibility of overlapping strains during the end and beginning of the study dates should be added to the limitations.
Author Response
Reviewer #1
Comments and Suggestions for Authors
Dear authors. I have enjoyed reading your manuscript. I consider a very interesting and relevant work. I send some comments and improvements.
Reply:
We thank you deeply for taking time and effort to review our manuscript. The quality of our manuscript has improved thanks to your comments.
Comment 1:
TITLE
Adequate, reports the correctness of the study.
Reply 1:
Thank you very much for the positive feedback.
Comment 2:
ABSTRACT
The first sentence is previewing the results of the study. It is suggested to replace with a sentence that gives context to the study.
Reply 2:
We have changed the sentence “The clinical characteristics of COVID-19 due to the Alpha and Delta SARS-CoV-2 variants differ.” to “Coronavirus infections occurred in repeated waves caused by different variants of severe acute respiratory syndrome coronavirus 2 (SARS-CoV-2), with the number of patients increasing during each wave.” (Lines: 9–11)
Comment 3:
INTRODUCTION
In the first paragraph, a context of the variants and the historical succession of the waves in the covid should be given. I consider this reference very relevant (not mine): Pecho-Silva S, Barboza JJ, Navarro-Solsol AC, Rodriguez-Morales AJ, Bonilla-Aldana K, Panduro-Correa V. SARS-CoV-2 Mutations and Variants: what do we know so far? Microbes Infect Chemother. 2021; 1: e1256.
Reply 3:
We have included the suggested reference. (Line 31)
Comment 4:
RESULTS
Very well presented.
Reply 4:
Once again, thank you very much for the kind review.
Comment 5:
DISCUSSION
The possibility of overlapping strains during the end and beginning of the study dates should be added to the limitations.
Reply 9:
We have added the sentence “Finally, there is a possibility of strains overlapping during the end of the Alpha and beginning of the Delta study periods.” in the limitations section as per your suggestion. (Lines 246–247)
Reviewer 2 Report
The present descriptive study compares the characteristics of patients requiring oxygen derived to Hospital-at-Home while waiting for hospitalization between the Alpha and Delta waves.
The authors acknowledge the difference in age between waves. These differences are reflected in other variables analyzed like presence of comorbidities and alcohol and smoking habits. However, no adjustments have been made for case fatality rate comparisons. The authors also acknowledge the age-effect of vaccination, but the vaccination status of the patients was not included in the study. Raw comparison of the case fatality rate is misleading when the Delta group is much younger than the Alpha group.
Minor comments:
Line 44 correct “awaining”.
Line 67 complete “severe…”.
Line 79 correct “servise”.
Lines 81-86 revise.
Table 2: no % are shown for the Japan Coma Scale.
Lines 192-198 revise.
Author Response
Reviewer #2
Comment 1:
The present descriptive study compares the characteristics of patients requiring oxygen derived to Hospital-at-Home while waiting for hospitalization between the Alpha and Delta waves.
Reply 1:
Thank you for carefully reviewing our manuscript and providing thoughtful suggestions and insights.
Comment 2:
The authors acknowledge the difference in age between waves. These differences are reflected in other variables analyzed like presence of comorbidities and alcohol and smoking habits. However, no adjustments have been made for case fatality rate comparisons. The authors also acknowledge the age-effect of vaccination, but the vaccination status of the patients was not included in the study. Raw comparison of the case fatality rate is misleading when the Delta group is much younger than the Alpha group.
Reply 2:
We have added the word “crude” before “case fatality rate” for clarification. In addition, we have added the following sentences in the limitation section:
“Fifth, to compare the case fatality rate of COVID-19 patients treated by the AHHC service between the Alpha and Delta waves, we could not adjust the confounder because the number of deaths was low. Sixth, we did not retrieve information on patients’ vaccination status; thus, we could not include this as a variable in the study.” (Lines 242–247)
Comment 3:
Minor comments:
Line 44 correct “awaining”.
Reply 3:
We have corrected “awaining” to “awaiting.” (Line 45)
Comment 4:
Line 67 complete “severe…”.
Reply 4:
We are sorry for the confusion. We have added numbering to classify COVID-19 severity into four groups. (Line 67–68)
Comment 5:
Line 79 correct “servise”.
Reply 5:
We have corrected the spelling of the word (Line 80).
Comment 6:
Lines 81-86 revise.
Reply 6:
We have revised the mentioned paragraph as per your comment from “After request a consultation from a public health center, AHHC provided patients with moderate I disease receive oxygen and three telephone consultations per day; and those with moderate II disease receive oxygen, dexamethasone, and eight telephone consultations per day In the Alpha and Delta wave, the Ministry of Health, Labour, and Welfare (MHLW) had not approved antiviral drugs (e.g., favipiravir or remdesivir) in out-of-hospitl setting.” to “After consultation by a public health center, patients with moderate I illness re-ceived oxygen and three telephone consultations per day from AHHC; those with moderate II severity received oxygen, dexamethasone, and eight telephone consulta-tions per day. During the Alpha and Delta waves, the Ministry of Health, Labour, and Welfare (MHLW) had not approved antiviral drugs (e.g., favipiravir or remdesivir) for out-of-hospital settings.”
Comment 7:
Table 2: no % are shown for the Japan Coma Scale.
Reply 7:
Thank you for your suggestion. We have added the prevalence (%) of people according to the Japan Coma Scale in Table 2.
Comment 8:
Lines 192-198 revise.
Reply 8:
We have accordingly revised the paragraph “In this study, no patient died during HaH care. In the Alpha and Delta wave, death at home became an issue, especially elderly in the Alpha wave and middle-aged in the Delata wave in Japan [26]. The reason was reported that the public health centers were too busy to observe at the waves [27]. In this study, the prefectures requested AHHC to provide HaH service, leading to the cooperation between the public health centers and the AHHC service, and no deaths at home among the patients. Therefore, HaH care may be safe and effective in patients with COVID-19.” to “In this study, no deaths occurred at home in patients treated by the AHHC service during the study periods. In general, deaths at home were a result of delayed intervention or delayed discovery of abnormality, especially in the elderly during the Alpha wave and the middle-aged in the Delta wave in Japan [26]. This was reportedly due to public health centers being too busy to observe patients at home during the waves [27]. Therefore, HaH care may be safe and effective in patients with COVID-19.”
(Lines 193–198)
In addition, we have made use of an English editing service. The certificate for English editing has been attached.
Round 2
Reviewer 2 Report
The authors could not address some of my comments, but they have acknowledged them as limitations.
Percentages of the category ‘Delirium’ of the Japan Coma Scale are not correct.
Author Response
Thank you for carefully reviewing our manuscript.
We have corrected the prevalence of the category ‘Delirium’ of the Japan Coma Scale in Table 2.